# Navigating the Labyrinth: Evaluating LLMs' Ability to Reason About Search Problems

**Nasim Borazjanizadeh**
*Berkeley AI Research, UC Berkeley*

**Roei Herzig**
*Berkeley AI Research, UC Berkeley*

**Trevor Darrell**
*Berkeley AI Research, UC Berkeley*

**Rogerio Feris**
*MIT–IBM Watson AI Lab*

**Leonid Karlinsky**
*MIT–IBM Watson AI Lab*

**Reviewed on OpenReview:** *https://openreview.net/forum?id=oub2I1ioL5*

## Abstract

Large Language Models (LLMs) have recently achieved impressive performance in math and reasoning benchmarks. However, they often struggle with logic problems and puzzles that are relatively easy for humans. To further investigate this, we introduce a new benchmark, SearchBench, which contains 11 unique search problems inspired by intuitive puzzles. Each SearchBench problem type is equipped with automated pipelines to generate an arbitrary number of instances and analyze the feasibility, correctness, and optimality of LLM-generated solutions. We show that using step-by-step, language-only reasoning, even the most advanced LLMs fail to solve SearchBench; for example, GPT-4 and o1-preview solve only 1.4% and 18.6% of problems, respectively. The reason is that SearchBench problems require considering multiple pathways and performing backtracking, posing a significant challenge to auto-regressive models. Interestingly, performance is significantly boosted when we prompt models to generate a complete A* search algorithm—a comparatively more cognitively difficult task for humans. This approach effectively offloads the iterative search and backtracking process from the models, which they struggle to perform in text. This in-context learning baseline is further enhanced via a Multi-Stage-Multi-Try (MSMT) inference method, increasing GPT-4's rate of correct solutions to over 57%.

## 1 Introduction

The advent of Large Language Models (LLMs) has revolutionized the field of natural language processing, with models such as Llama3.1 (Meta, 2024), GPT-4 (OpenAI, 2023), and o1-preview (OpenAI, 2024) demonstrating unprecedented performance on math and science QA benchmarks, such as GSM8k (Cobbe et al., 2021) and GPQA (Rein et al., 2023). However, LLMs still exhibit surprising failures on some intuitive tasks (Bian et al., 2023; Qin et al., 2023; Marcus, 2020) and struggle with multi-step compositional reasoning, combinatorial problems, and planning (Dziri et al., 2023; Valmeekam et al., 2022; Wu et al., 2023). The problems of SearchBench are inspired by popular puzzles, are predominantly NP-hard combinatorial problems, and necessitate exploring multiple action paths and backtracking to previous states. Here, NP-hardness refers to the generalized problem families rather than to the fixed finite instances; the empirical

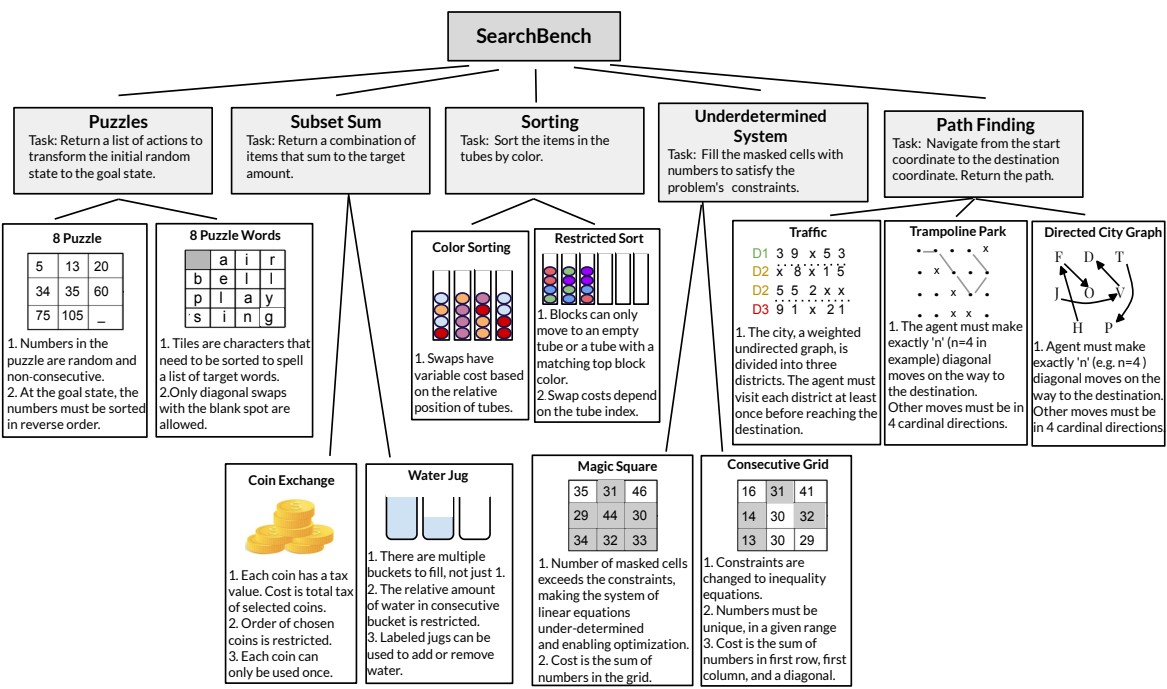

Figure 1: The taxonomy of SearchBench. The five nodes in level one represent the problem categories, and the 11 nodes in level two represent the problem types. We detail how the rules of known puzzles and combinatorial problems are modified in SearchBench to ensure LLMs haven't encountered solved instances of the problems during training.

difficulty of the fixed instances is characterized by solution depth (i.e., number of actions needed to solve the problem), maximum search tree expansion, size of the feasible state-space, and runtime of the reference A* algorithm.

SearchBench is challenging for LLMs due to several factors. The autoregressive architecture of current LLMs forces solving problems sequentially, which makes tasks requiring backtracking challenging (Dziri et al., 2023). Additionally, natural language is inherently linear and inefficient for representing complex relational structures (Dziri et al., 2023; Borazjanizadeh & Piantadosi, 2024). Moreover, in combinatorial problems, the number of possible states increases exponentially with the number of actions, making naive exploration of the state space ineffective. Our empirical results show that even the most capable models solve less than 20% of SearchBench problems end-to-end. To successfully solve SearchBench, a model must backtrack, consider multiple reasoning paths, and identify the most optimal outcome among many feasible options. These capabilities are essential for robust reasoning, making SearchBench a valuable benchmark for evaluating LLMs' reasoning capabilities as they continue to evolve.

SearchBench has five problem categories: (i) pathfinding, (ii) puzzles, (iii) subset sum, (iv) sorting, and (v) under-determined systems; further divided into 11 unique problem types. Each problem type is inspired by known puzzles and combinatorial problems, but includes modified rules to ensure they differ substantially from solved instances of the original problems that appear on the internet and are likely observed by LLMs during their pre-training. We generate about 100 instances of varying difficulty per problem type using an automatic pipeline, totaling 1,107 fixed instances. Each problem type in SearchBench also includes an automatic evaluation pipeline that assesses LLM-generated solutions on three dimensions: feasibility (choosing sequence of actions that adhere to the problem's rules), correctness (achieving the goal state), and optimality (finding the least cost solution).

Our results show a contrast between two computational settings. While models fail at the cognitively simple task of solving these puzzles step-by-step, we show their performance is significantly boosted when prompted to generate a complete A* search algorithm. In text-only prompting, the model must generate and maintain

the action sequence within a single autoregressive trace. In A* prompting, the prompt supplies a search prior and the generated Python program performs state expansion and backtracking at execution time. This approach succeeds because it leverages the model's strength in code generation, which is not an iterative task, offloading the exploration of a large action space from LLMs to code execution, resulting in improved performance (Fig. 3). A* itself is a heuristic-based search algorithm with optimality guarantees under standard admissibility and consistency conditions (Hart et al., 1968); compared with uninformed breath-first search (BFS) and depth-first search (DFS), it uses a problem-specific heuristic to prioritize states, while BFS has high memory requirements and DFS does not generally guarantee a minimum-cost solution (Russell & Norvig, 2020).

Additionally, to improve the quality of the generated A* codes, motivated by recent work showing that multiple inferences and task decomposition improve LLM performance (Wang et al., 2022; Yao et al., 2023a; Long, 2023), we introduce the Multi-Stage-Multi-Try (MSMT) inference strategy. In this approach, we divide code generation into two stages. First, we prompt the model to write an instance-agnostic A* algorithm for the problem type. We then verify this implementation against a set of unit tests, without evaluating the solution, to check if (i) the code is executable, (ii) returns a list as output, and (iii) the data type of list elements is correct. Second, we instruct the model to implement the 'initialize' function, which encodes the variables specific to each problem instance. Our MSMT A* method (Fig. 2) significantly improves LLMs' ability to solve search problems, outperforming other prompting strategies we used, including 0-shot text, 4-shot Chain-of-Thought (CoT) (Wei et al., 2022) text, 0-shot code, and 4-shot A* prompting with naive greedy decoding. This method constitutes our strongest baseline on SearchBench; under the respective inference procedures evaluated here, GPT-4 prompted with MSMT A* surpasses the o1-preview model. However, even using this approach, there remains considerable room for improvement on SearchBench, underscoring the challenge it presents in advancing LLMs' reasoning capabilities.

To summarize, our main contributions are as follows: (i) We introduce the SearchBench benchmark, designed to evaluate LLMs' ability to solve combinatorial problems that require search and backtracking. (ii) We demonstrate a key LLM bottleneck by showing that while models fail at iterative reasoning in text, they can successfully generate complex search algorithms; we then harness this capability with the MSMT A* framework to boost LLMs' performance on SearchBench. (iii) We use SearchBench to thoroughly analyze the reasoning capabilities of several LLMs, including GPT-4 and o1-preview, by employing various prompting and inference methods. This analysis uncovers the limitations of LLMs in tasks requiring iterative reasoning and highlights the potential for improving LLM performance on SearchBench. The code, dataset, prompts, and evaluation scripts are available at `https://github.com/NasimBorazjani/Navigating_Labyrinth`.

## 2   SearchBench Benchmark

SearchBench includes five categories of problems: puzzles, subset sum, sorting, pathfinding, and under-determined systems. Generally, our problems involve an initial state, a goal state, and a set of possible actions, and the task is to find a sequence of actions from the initial to the goal state with minimum cost. In theoretical computer science, combinatorial problems are classified into four types: existence, construction, enumeration, and optimization problems (Wilson, 2016). These types characterize the form of the requested solution (instance difficulty is characterized separately, based on the size of the search space). To ensure broad representation, we selected at least one problem category from each of these types for SearchBench. To ensure broad representation, we selected one problem category from each of these types for SearchBench. Particularly, subset sum problems represent the existence category, where the task is to determine if a subset of a given set sums to a specified value (refer to Tab. 1 for an example problem in this category). The 8-puzzle and 8-puzzle words fall under construction problems, which involve solving puzzles. Sorting problems, such as color sort and restricted sorting, are enumeration problems. Pathfinding problems are categorized as optimization problems.

Additionally, we introduce a new category of NP-hard combinatorial problems in SearchBench, under-determined system problems. These problems consist of constraint satisfaction problems which are solved by defining a system of linear equations, and do not require search over states. We modified them to include fewer constraints than unknown variables, allowing for multiple correct solutions, and defined a unique cost

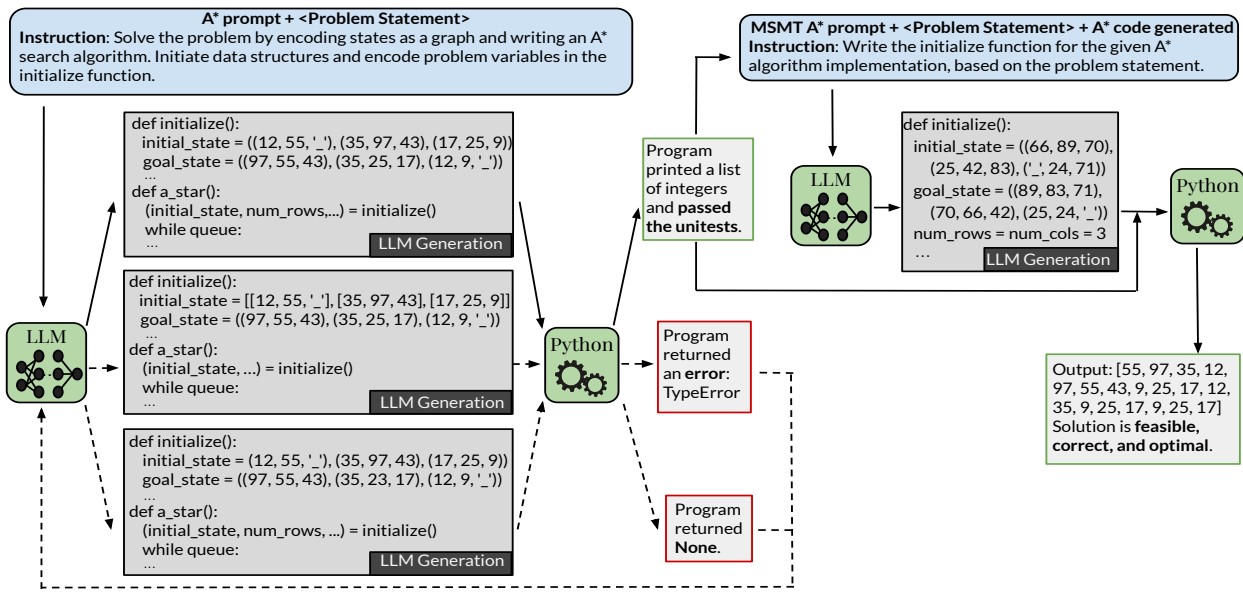

Figure 2: Our Multi-Stage-Multi-Try (MSMT) A* prompting approach.

Table 1: The example below shows an instance of the natural-language problem statement supplied to the model. Green indicates instance-specific components, and orange highlights the modified rules specific to SearchBench. GPT4 fails to generate a feasible solution for this instance using the three baseline prompting methods, but produces a correct, non-optimal solution using A* and MSMT A*.

Table 1 shows the exact natural-language problem statement supplied to the model, with instance-specific components in green and SearchBench-specific rule modifications in orange; the full prompts for all five methods are provided in the supplementary material.

| Problem statement |
| --- |
| In the 'taxed coin exchange' problem, you are required to choose a subset of coins from this list [3, 6, 9, 10, 13, 15, 18, 5, 21, 19, 12, 15, 5, 9, 4, 16, 8, 4, 7, 7, 7, 2, 16, 14, 18, 3, 89, 21, 12, 10, 7, 14, 4, 11, 6, 20], such that the sum of the chosen coins adds up to 229. Each coin occurrence in the list can only be used once. Also coins carry a tax value. The tax values for each coin is 14: 1, 89: 13, 2: 2, 5: 2, 4: 4, 6: 6, 8: 2, 16: 5, 21: 4, 20: 2, 18: 9, 11: 10, 10: 3, 12: 12, 15: 5, 13: 1, 3: 1, 19: 19, 7: 7, 9: 3, where the tax for coins of the same value is the same. Also, if the coin chosen is smaller than the previous one, it must have an even value, otherwise, if the coin is larger than or equal to the previous coin chosen, it must have an odd value. The objective is to determine which subset of coins should be selected to minimize the total tax paid. The solution should be presented as a list of numbers, representing the value of the coins chosen in order, with the first coins chosen being in index 0, formatted in Python syntax. |

function to enable search for a minimum-cost solution. This category was added in order to evaluate models' ability to generalize to novel combinatorial problems.

We selected 2-3 problem types for each category, resulting in 11 total problem types. Each type has a unique state space. For example, in 8-puzzle words, each state is an $n \times m$ table of characters, while in coin exchange, each state is an ordered subset of given coins. We modified the rules to ensure that solved instances of SearchBench were not encountered during the LLMs' massive internet-scale training. The SearchBench taxonomy and rule modifications are illustrated in Fig. 1.

To construct SearchBench, we implemented an automatic generation pipeline for each problem type, ensuring each generated instance is solvable. We generated approximately 100 instances per type, resulting in a total of 1107 problem instances. The benchmark is then fixed. The generation pipelines can create instances with adjustable difficulty levels. The resulting difficulty scores are relative within each problem type. Difficulty is defined by the state space size of the instance (see Tables 4 and 5 for example solution depth, maximum successor count, state-representation size, and reference A* runtime).

In SearchBench instances with minimum difficulty require a few actions and maximum difficulty is set such that problems could be solved correctly but not optimally by humans (See Appendix Sec. G for an analysis of the search space size). Hence, maximum human correctness on SearchBench could be considered approximately 100%; this is a qualitative estimate based on the authors' experience, not a controlled human study. Moreover, prior studies show that humans can solve classic puzzle and combinatorial-optimization tasks related to SearchBench, although their solutions can be suboptimal and their performance declines as the size of the state space increases (Pizlo & Li, 2005; Chronicle et al., 2006; Atwood & Polson, 1976).

In contrast to other reasoning benchmarks (Cobbe et al., 2021; Hendrycks et al., 2021; Patel et al., 2021; Clark et al., 2020; Tafjord et al., 2020; Sap et al., 2019; Le et al., 2019) that only measure correctness, to gain a more comprehensive understanding of LLM performance on SearchBench, our evaluation pipeline assesses LLM solutions across 3 dimensions: Feasibility, Correctness, and Optimality. Feasibility determines if any of the actions chosen violate the problem rules (e.g. passing through labyrinth walls). Correctness requires that the solution is both feasible and reaches the goal state from the given start state. Optimality indicates that the solution is both correct and has the minimum cost w.r.t. known optimum. For each SearchBench problem, we implemented a fast A* algorithm with a provably admissible and consistent heuristic, to produce the optimal solution. We ran this implementation for each instance in the benchmark to obtain its optimal solution.

We note that even though correctness implies feasibility, and optimality implies correctness, feasibility and correctness are valuable intermediate metrics in determining how close the models are to generating the fully correct solution. For example, in traffic problems, GPT4 often fails to record the first city visited, resulting in a feasible but incorrect solution. Defining feasibility helps distinguish this mostly correct implementation from more erroneous solutions. Correctness is stricter than feasibility and indicates that search-related tasks were implemented correctly, but the heuristic or recorded cost is incorrect, leading to non-optimal solutions.

## 3  Evaluated Methods

We use three baseline prompting methods to evaluate LLMs on SearchBench: 0-shot text, 4-shot CoT text, and 0-shot code. Additionally, we use two new code-based methods: 4-shot A* prompting and MSMT A*. See supplementary material for full prompts for each of the five approaches and GPT4's generated solution.

To ensure the generality of our prompting methods, we selected one in-context example from each of the four SearchBench categories that are different from the category of the evaluated problem. This minimizes the similarity between the rules and context of the solved examples and the evaluated problem, and tests whether the model can solve unrelated combinatorial problems. Thus, if a model finds an optimal solution using these methods, it demonstrates true generalization rather than prompt-specific improvements. In Sec. 6, we further analyze the impact of including an example from the same top-level problem category. Additionally, 4-shot is the upper limit on the number of in-context examples due to the models' context length limit. For an analysis of the effect of fewer demonstrations (shots) on performance, see Appendix Sec. B.

**0-shot text and 4-shot CoT text prompting methods:** In the text-based prompting methods, we instruct the model to solve the problem in an end-to-end manner, using text only. In 4-shot CoT prompts, the in-context examples include a representation of the intermediate states drawn using ASCII characters after each action to prevent hallucinations and illogical leaps in reasoning.

**0-shot code prompting method:** This method instructs the LLM to produce a Python code that solves the given problem. The generated code is then executed to derive the final answer.

**A\* Prompting:** In this approach, we prompt the LLM to implement an A* algorithm that solves $\mathcal{P}_i^C$ - a problem instance number $i$ of problem category $C$, providing four in-context examples of A* codes for four problems $\mathcal{P}_j^{\hat{C}}$, each selected from a different category $\hat{C} \neq C$. To implement A* for the target SearchBench problem, the LLM must perform abstract reasoning to devise a search strategy applicable to any state within the search space. This contrasts with solving problems end-to-end in text, where the model has access to the variables of each state, eliminating the need for abstract reasoning. However, end-to-end approaches require

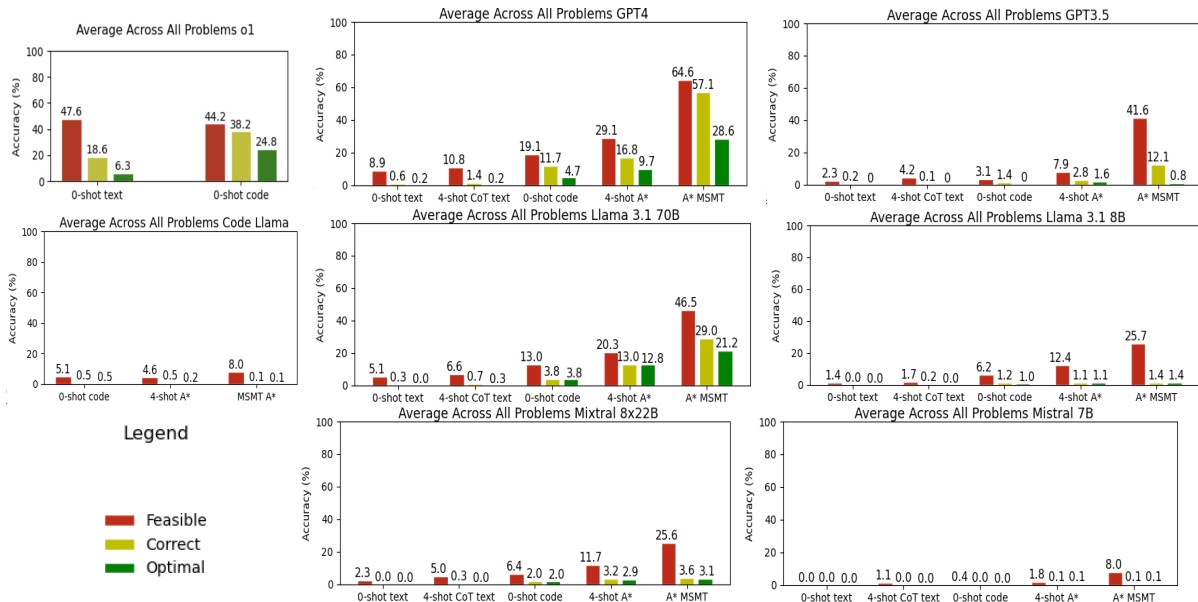

Figure 3: Average rate of feasible, correct, and optimal solutions for all problems using o1, GPT4, GPT3.5, Code Llama, Llama 3.1 70B, Llama 3.1 8B, Mixtral 8x22B, and Mistral 7B.

the model to perform every step of the iterative computations involved in searching the state space. In this method, the model must formulate the problem-specific search strategy, while code execution performs the iterative state-space exploration.

The in-context examples include detailed comments before each code segment, explaining the reasoning used to develop the strategy implemented within the code segment. These comments serve as CoT reasoning for devising the search strategy implemented in the code.

**Multi-Stage-Multi-Try (MSMT) A\* Prompting:** In this method, the model receives the same in-context examples as the 'A\* prompting', with different instructions. Here, inference is done in two stages as demonstrated in Fig. 2. In the first stage, the model is instructed to implement the code as two functions: the 'a_star' function includes an instance-agnostic A\* algorithm for the target problem type, and the 'initialize' function encodes the variables given in the problem statement. We then verify if the generated code satisfies the following set of unit tests: (i) code is executable; (ii) code returns a list; (iii) and the list elements match the data type specified by the problem statement. If the code fails any unit test, MSMT re-generate the code. Next, in the second stage, the LLM is instructed to implement an 'initialize' function, conditioned on the verified 'a_star' function from stage 1 for each instance of the problem type. The inclusion of simple unit tests, which can be expanded to more detailed tests if needed, offers a robust method for filtering out erroneous samples from the model's generations.

In our MSMT A\* prompting approach, the model generates the full A\* algorithm end-to-end without feedback on the correctness of target-instance solutions, similar to how text-based prompting methods operate. Importantly, our MSMT A\* does not rely on the majority vote of multiple solutions. Instead, the solution returned by the first model-generated code that passes the unit tests is taken as the final answer. This results in increased efficiency of MSMT A\*, requiring only 1.5x number of generations per problem on average, compared with 5–100 samples in majority-vote approaches (Wang et al., 2022).

## 4 Related Work

**Mathematical and Reasoning Benchmarks**: Evaluating LLMs (Brown et al., 2020; OpenAI, 2023; 2022; Chung et al., 2024; Chowdhery et al., 2023; Rae et al., 2021; Taylor et al., 2022; Thoppilan et al., 2022) on mathematical and reasoning tasks has been a focus of recent research in natural language processing,

leading to the development of benchmarks such as BIG-BENCH (Srivastava et al., 2022), GSM8K (Cobbe et al., 2021), AQUA (Ling et al., 2017), SVAMP (Patel et al., 2021), CommonsenseQA (Talmor et al., 2018), StrategyQA (Geva et al., 2021), and MATH (Hendrycks et al., 2021). However, these benchmarks have limitations. GSM8K problems are relatively simple and often require a repetitive reasoning pattern to solve, and problems in BIG-BENCH are mostly single-step reasoning tasks. The MATH dataset, while more challenging, may not accurately assess a model's generalizable reasoning capabilities due to the advanced mathematical skills required. When prompted to solve problems using CoT prompting in text, LLMs perform well on these tasks; however, they fail on SearchBench's problems, indicating that these benchmarks offer limited insight into LLMs' ability to systematically explore a state space.

**Application of LLMs to Combinatorial Problems**: Few studies such as (Yang et al., 2023; Liu et al., 2024; Masoud et al., 2024) have explored solving select combinatorial problems like the Traveling Salesman Problem with LLMs. (Mittal et al., 2024) introduced a dataset of combinatorial problems, "PuzzleBench". The problems of this dataset are selected to be representable in a symbolic solver (SMT2.0) and have hand-engineered symbolic representations for input states and outputs, which can limit the generalization of this dataset. Similarly, problems selected by (Mittal et al., 2024) and (Iklassov et al., 2024) are instances of the classical combinatorial problems, raising issues of memorization, as algorithm implementations for instances of such problems are available online.

SearchBench stands out in several ways (i) Generalizability: SearchBench problems are described only in natural language, with no restrictions on rules or actions, and cover a wide variety of combinatorial problems, ensuring that a model capable of solving SearchBench can generalize to other combinatorial problems. (ii) Uniquely Modified Rules: This prevents memorization, as algorithms for classic versions of the problem are available online. (iii) Optimal Solution: Each problem type has a uniquely defined cost, enabling evaluation against a minimum cost solution. (iv) Multi-Dimensional Evaluation: This provides deeper insights into how close models are to deriving an optimal solution. (v) Automated Instance Generation: This enables difficulty control and reduces contamination risk, as new instances can be generated on demand.

**Prompting Strategies**: Sophisticated prompting strategies have been developed to enhance models' reasoning abilities. One notable approach is Chain-of-Thought (CoT) prompting (Wei et al., 2022), which prompts LLMs to generate intermediate steps leading to the final output. This technique has led to advanced variations, including Tree-of-Thoughts (Yao et al., 2023a; Long, 2023), and Graph-of-Thought (Yao et al., 2023c; Lei et al., 2023; Besta et al., 2024) methods that maintain a tree of intermediate generations. However, these methods rely on evaluating and rejecting intermediate steps, which does not integrate well with our problems. In search problems, intermediate states can't be easily classified as correct or incorrect, and all possible actions must be considered to find the optimal solution. Additionally, the state space of combinatorial problems grows exponentially, making it impractical for LLMs to navigate the frontier of the search tree without incorrectly disregarding most feasible states.

Other prompting methods, such as Decomposition strategies (Khot et al., 2022; Zhou et al., 2022; Zhang et al., 2023), simplify complex tasks into smaller, manageable subtasks to improve performance. Additionally, systems like LLM-Augmenter (Peng et al., 2023) rely on external databases to verify segments of the LLM's output. In this work, we propose the A* prompting strategy, where we prompt the model to solve problems by implementing a unique A* algorithm. Similarly, our A* MSMT approach decomposes the task of implementing the search algorithm into two stages and checks the model's generations against external validators; we use simple unit tests instead of external data sources or solved solution instances.

Program-aided approaches such as PAL similarly offload execution to a runtime (Gao et al., 2023), while ReAct interleaves language-model reasoning with external actions (Yao et al., 2023b). These settings are complementary: MSMT is a pragmatic baseline for evaluating SearchBench; the benchmark and systematic analysis are the paper's primary contribution.

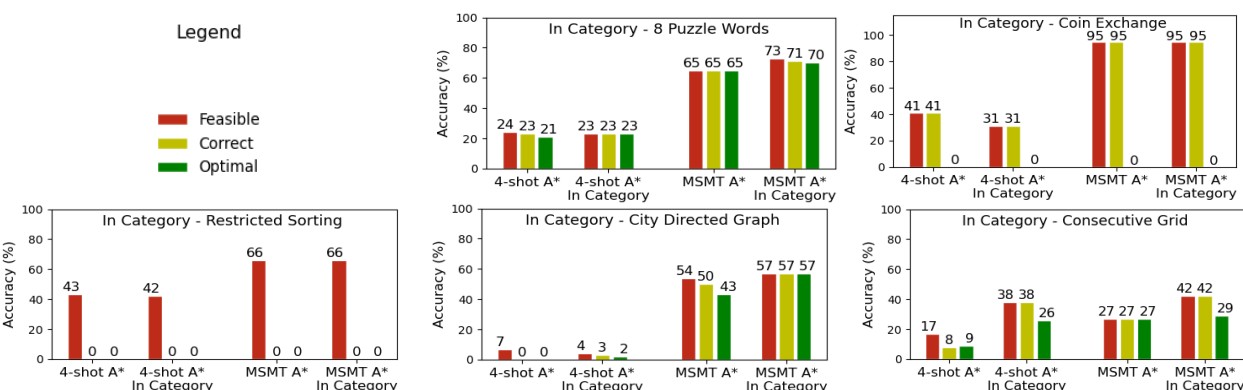

Figure 4: Comparing GPT4's performance, using A* prompting approaches, when one of the in-context examples is switched to a problem that shares the same category as the inference problem.

## 5 Experiments

We evaluated the performance of GPT4, GPT-3.5, and Code Llama Instruct 34B (Roziere et al., 2023) [1], Llama 3.1 70B, Llama 3.1 8B, Mixtral 8x22B (Mistral, 2023b), and Mistral 7B (Mistral, 2023a) on SearchBench, using the following five prompting methods described in Sec. 3: 0-shot text, 4-shot CoT text, 0-shot code, 4-shot A*, and 4-shot MSMT A*. Results are summarized in Fig. 3.

**Implementation details:** We used GPT4, GPT-3.5 Turbo (GPT3.5 hereafter), and o1-preview (o1 hereafter) via OpenAI platform APIs. All code evaluations were conducted on a machine with 96 64-bit Intel Xeon Gold 5220R CPUs, a maximum speed of 4GHz, and a 71.5 MiB Level 3 cache.

**0-shot text and 4-shot CoT text prompting methods:** As shown in Fig. 3, the correct solutions rate is below 1% for all of the models using 0-shot text prompting, and less than 9% of GPT4 solutions are feasible (follow the problem rules) using this method. This is expected as the exponentially growing state space size of SearchBench problems and the difficulty of backtracking during auto-regressive generation make it challenging to solve SearchBench problems using text-based prompting, even with the strongest LLMs. Moreover, 4-shot CoT text prompting only improves the rate of feasible solutions generated by less than 3% for all models. This shows that the inherent complexity of search problems from SearchBench cannot be effectively addressed by text-based prompting alone.

Finally, we evaluate the recent o1 model (OpenAI, 2024), which is designed for comprehensive reasoning. As shown, this model struggles with SearchBench problems, solving less than 19% correctly with 0-shot text; however, it significantly outperformed other models in solving the problems end-to-end. We did not evaluate this model with A* prompting and MSMT A* due to its limited context length. Additionally, o1 reasons over various solutions internally; the comparison with GPT-4+MSMT therefore contrasts two different inference-time compute scaling methods: extended reasoning tokens versus multiple-try inference.

**0-shot code prompting method:** This prompting method improves performance over text-based prompting for all models except Mistral 7B, which remained close to 0%. This is expected, as using Python to compute intermediate steps and execute the iterations of the algorithms devised by the LLMs reduces the load on the models. As seen in Fig. 3, o1 solved 38.2% of the problems correctly, 19.1% of GPT4's code generations result in a feasible solution, with only 11.7% being correct. The next best performance was achieved by Llama 3.1 70B, which solved 13% of the problems correctly. For an analysis of the computation time of programs generated by the LLMs, please refer to Appendix Sec. E.

**A* Prompting:** As shown in Fig. 3, A* prompting improves the performance of all models on SearchBench except for Code Llama, which shows almost no improvement, indicating potential limitations of this model in in-context learning or following the given instructions. GPT-4's rates of feasible, correct, and optimal solutions increase by 10%, 5%, and 5%, respectively, and Llama 3.1 70B's rates increase by 7%, 9%, and 9%.

---

[1]Finetuned on the Phind dataset (Phind, 2023)

**MSMT A\*:** As shown in Fig. 3, MSMT A\* prompting significantly boosts the performance of all models. Using this method, GPT4 correctly solved 57.1% of SearchBench problems and achieved a 28.6% rate of optimal solutions, outperforming o1. GPT4's performance improved consistently across all problem types compared to other prompting strategies (see Appendix Sec. C for a detailed analysis of performance on each problem type). Other LLMs also showed strong improvements, except for Code Llama, which only improved in feasibility because it still struggled to follow the instructions. However, the 28.6% optimal performance of GPT4 using MSMT A\*, although inspiring, still leaves room for further improvements, underlining the importance of SearchBench for future research.

## 6  Ablations and Analysis

Here we provide a comprehensive analysis of SearchBench using GPT4. For further analysis, please refer to Appendix Sec. B, C, D, E, and G.

**Does including a more similar problem in prompt improve GPT4's performance?** In our main experiments with A\* and MSMT A\* (Fig. 3), we used four in-context examples, each from a different category than the target problem (Sec. 3). This ensured that no exact segment of the target solution was included in the prompt, hence better measuring LLM's reasoning generalization. Here, we evaluated GPT4's performance when a solved example from the same category but a different type as the evaluated problem, is included in the prompt. Results are summarized in Fig. 4. We observed small improvements, with up to 15 additional instances solved. This indicates that SearchBench problems within the same category still differ significantly in rules, constraints, and target A\* algorithm implementations.

The most significant improvement was observed for the Consecutive Grid problems from the under-determined systems category which involve filling in masked numbers according to constraints on the order of integers in a table. This category differs more significantly from other combinatorial problems, showing that including similar problems in the prompt leads to greater improvement for new tasks.

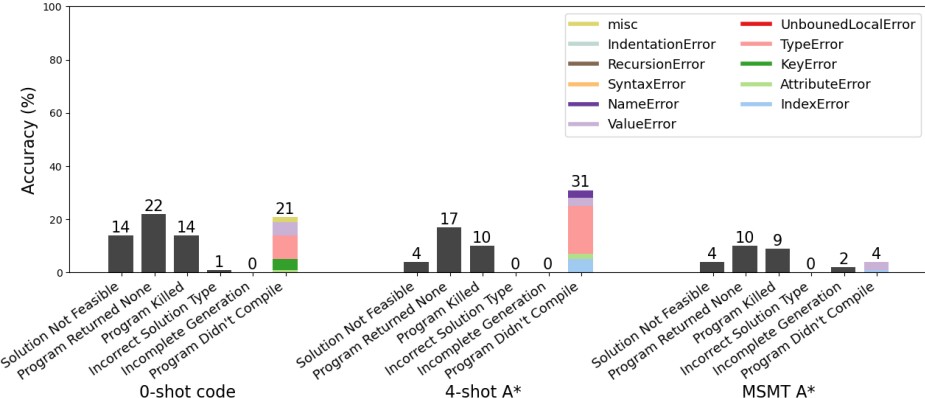

Figure 5: Rate of errors returned by python programs generated by GPT4, categorized into 6 error types, calculated across all SearchBench problems with an infeasible solution.

**What types of runtime errors occur, and how often, when executing GPT4's implementations?** We analyzed the errors returned by GPT4's generated codes that resulted in infeasible solutions. The results are shown in Fig. 5. For an analysis of correct versus optimal solutions see Appendix Sec. C and D.

Below, we categorized errors into six types: (i) Solution Not Feasible: code executed but returned an infeasible solution; (ii) Program Returned None: program failed to find a solution; (iii) Program Killed: program did not finish within the allotted time; (iv) Incorrect Solution Type: returned solution had the wrong data type; (v) Incomplete Generation: model ran out of tokens; and (vi) Program Didn't Compile.

As shown in Fig. 5, prompting the model with the A\* method results in more non-compiling implementations compared to 0-shot code prompting. This is expected as A\* is more complex and longer compared to the

simpler algorithms typically implemented by the model using 0-shot code prompting, such as the greedy algorithm, BFS, or DFS. However, the number of infeasible solutions decreases with A* prompting, indicating that the model can better reason about the problem using this method. Comparing A* prompting to the MSMT A* method, we observe a significant decrease in errors that fail at least one unit test, such as 'Program Returned None', 'Program Killed', 'Incorrect Solution Type', 'Incomplete Generation', and 'Program Didn't Compile'.

**What are the most common reasoning errors made in GPT4's A* implementations?** We manually analyzed 50 A* codes generated by GPT4 that returned non-optimal solutions across five problem types: three pathfinding problems and two puzzle problems. These problems were chosen because GPT4 showed the least and greatest performance improvement, respectively, using A* prompting compared to 0-shot code (see Appendix Sec. 3).

We identified seven distinct failure modes in the GPT4-generated A* implementations, each corresponding to a critical sub-task within the overall search strategy, where failing any subtask results in a suboptimal solution. The results are summarized in Table 2, showing the percentage of 'correct reasoning' for each subtask, excluding coding errors. As shown, in pathfinding problems, the most common mistake was not recording the list of visited coordinates, a 13% success rate, with the model often omitting the start coordinate in the path, leading to feasible but incorrect solutions. In puzzle problems, the frequent error was encoding the goal state, likely because our puzzles have unique goal states unlike the conventional 8-puzzle problem.

Table 2: The average accuracy of GPT4 on identified A* subtasks (failure modes) was analyzed using 50 code implementations for pathfinding and puzzle problems.

|                                     | Pathfinding Problems | Puzzle Problems |
| ----------------------------------- | :------------------: | :-------------: |
| Encoding Initial State              | 47%                  | 100%            |
| Encoding Goal State                 | 74%                  | 20%             |
| Recording the Path                  | 13%                  | 70%             |
| Exit Condition                      | 70%                  | 100%            |
| Iterating Successor States          | 57%                  | 100%            |
| Generating New State                | 87%                  | 100%            |
| Admissible & Consistent Heuristic   | 93%                  | 60%             |

## 7 Conclusion

In this work, we introduced SearchBench, a pioneering benchmark designed to evaluate LLMs' ability to solve new combinatorial search problems that require backtracking and considering multiple action sequences. We assessed LLMs using various text-based and code-based prompting methods and showed that while models fail at solving these puzzles step by step in natural language, their performance improves considerably when asked to implement an A* search algorithm—a cognitively more challenging task for humans but one that shifts the state space exploration burden from the model to code execution. This contrast reveals a key bottleneck in iterative reasoning and backtracking within autoregressive text generation, while also highlighting how their strengths in formalizing the search strategy in code, can be harnessed to overcome these limitations, where external execution performs the iterative state-space exploration. For broader impact, see Appendix Sec. H.

## 8 Limitations

The primary challenge in developing SearchBench was scaling the number of problem types. Designing unique search problems and creating pipelines to generate numerous instances with guaranteed solutions is both time-consuming and complex. Additionally, implementing a fast, instance-agnostic A* algorithm and developing evaluation pipelines to assess LLM-proposed solutions on multiple criteria further adds to the complexity. We did not conduct a controlled human study, so our discussion of human performance

is qualitative. Moreover, MSMT uses additional tries and unit-test filtering, so its comparison with single-generation methods and o1 concerns different inference-time compute-scaling strategies rather than identical model-call procedures.

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

## A   Appendix

## B   n-shot Ablation Experiments

To examine the effect of different numbers of demonstrations on GPT4's performance using A* and MSMT A* prompting methods, we performed ablation experiments with 2-shot and 3-shot A* prompts. 4-shot is the upper limit on the number of in-context examples due to the context length constraints of the models, including GPT4. In all few-shot experiments, the examples used in the prompts were not from the evaluated problem category. The results, summarized in Fig. 6, show a consistent trend of performance improvement with the addition of more examples, as expected.

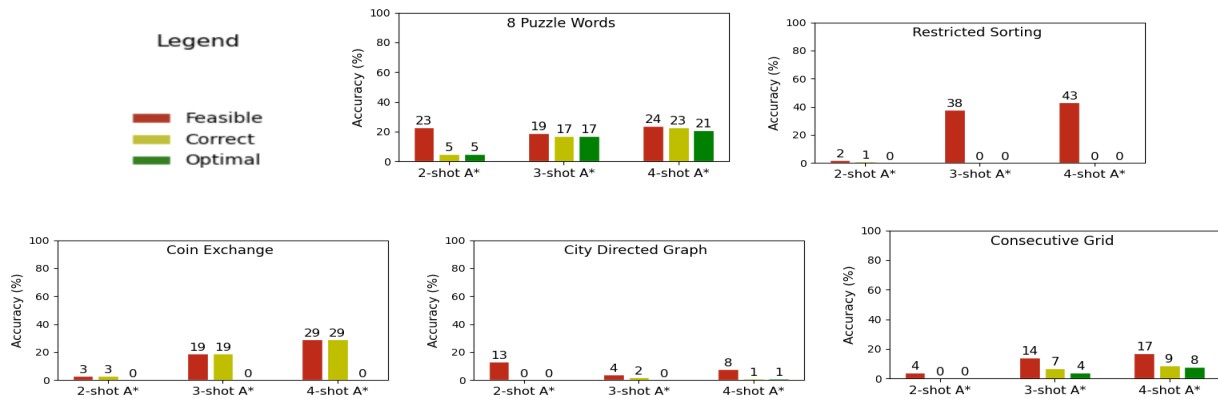

Figure 6: Comparing 2-shot, 3-shot and 4-shot performance of GPT4 between A*-prompting approaches.

## C   Detailed analysis of GPT4's performance on SearchBench

Tab. 3 details GPT4 code-based method performance for each of SearchBench's 11 problems. Consistently 4-shot A* prompting outperforms 0-shot code for most problems. Interestingly for problems in the pathfinding category, prompting GPT4 with 0-shot code outperforms A* prompting.

Examining closer, GPT4 mainly uses DFS for pathfinding in 0-shot code. While simpler than A*, DFS doesn't guarantee optimal solutions, as reflected in GPT4's high feasible and correct rates but lower optimal rates. Implementing A* with an admissible and consistent heuristic requires the model to implement a more complex strategy in the code involving additional constraints and more sophisticated data structures. This increases the likelihood of reasoning or coding errors, which could explain the dip in GPT4's performance using A* prompting compared to 0-shot code when solving these problems.

Figure 7 further analyzes the relationship between problem difficulty (quantified by state space size of the problem) and the performance of GPT4. As observed, the model's performance is generally higher on easier problems, particularly in terms of the rate of correct solutions. This is expected, as easier problems have a smaller state space to explore. However, the performance of the model does not change drastically across different difficulty levels. This indicates that the combinatorial problems in SearchBench are intrinsically hard for LLMs to solve in text due to the requirement for backtracking. Moreover, the difference in implementing an A* search algorithm for a difficult or easy instance of SearchBench is limited to encoding the initial and goal states. The rest of the algorithm implementation task remains the same. This is the reason why the model's performance is comparable across different difficulty levels, both using text-based and code-based methods.

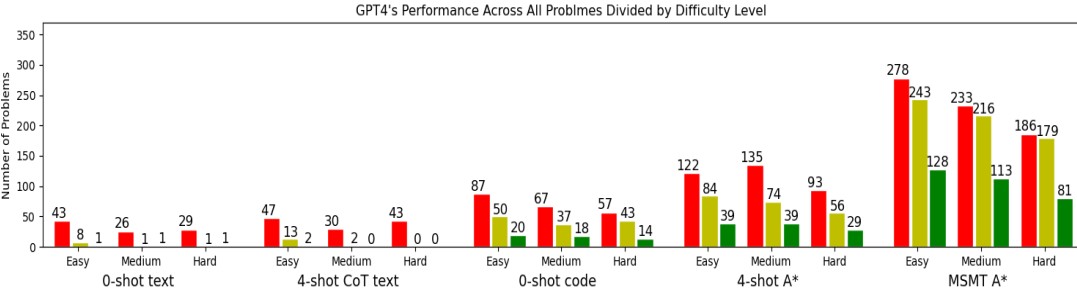

Figure 7: Count of feasible, correct, and optimal solutions generated by GPT4 via code-based methods for 3 levels of problem difficulty.

Table 3: GPT4's performance when prompted with our code-based approaches, on each problem type. The values are percentages of the feasible (F), correct (C), and optimal (O) solutions.

| Problem | 0-shot code | | | 4-shot A* | | | MSMT A* | | |
|---|---|---|---|---|---|---|---|---|---|
| 8 Puzzle | F: 3 | C: 0 | O: 0 | F: 63 | C: 60 | O: 60 | F: 76 | C: 68 | O: 68 |
| 8 Puzzle Words | F: 5 | C: 5 | O: 5 | F: 24 | C: 23 | O: 21 | F: 66 | C: 65 | O: 65 |
| Color Sorting | F: 17 | C: 1 | O: 1 | F: 41 | C: 35 | O: 6 | F: 91 | C: 91 | O: 0 |
| Restricted Sorting | F: 32 | C: 0 | O: 0 | F: 43 | C: 0 | O: 0 | F: 66 | C: 0 | O: 0 |
| Water Jug | F: 7 | C: 7 | O: 6 | F: 8 | C: 8 | O: 0 | F: 95 | C: 95 | O: 0 |
| Coin Exchange | F: 2 | C: 1 | O: 0 | F: 31 | C: 31 | O: 0 | F: 95 | C: 95 | O: 0 |
| Traffic | F: 65 | C: 50 | O: 13 | F: 24 | C: 5 | O: 5 | F: 65 | C: 60 | O: 60 |
| Trampoline Matrix | F: 27 | C: 27 | O: 22 | F: 51 | C: 4 | O: 4 | F: 57 | C: 53 | O: 46 |
| City Directed Graph | F: 29 | C: 28 | O: 1 | F: 7 | C: 0 | O: 0 | F: 55 | C: 51 | O: 45 |
| Magic Square | F: 3 | C: 1 | O: 0 | F: 8 | C: 5 | O: 0 | F: 14 | C: 14 | O: 0 |
| Consecutive Grid | F: 15 | C: 2 | O: 0 | F: 17 | C: 9 | O: 8 | F: 27 | C: 27 | O: 27 |

# D   Examples of Correct but Non-optimal Solutions

The following examples illustrate the correctness–optimality gaps in Table 3.

## D.1   Color Sorting

**Problem statement.** The instance contains three tubes, each initially holding four colored balls. Only the topmost ball of a tube can be moved, and it must be placed on top of another tube. Each tube has capacity six. The goal is to reach any arrangement in which every tube contains balls of only one color. A solution is represented as a list of tuples (`i`, `j`), indicating a move from tube $i$ to tube $j$, with tube indices starting from zero. The objective is to minimize the number of moves.

```
initial_state = (
    ('Green', 'Red', 'Green', 'Red'),
    ('Blue', 'Blue', 'Red', 'Green'),
    ('Red', 'Blue', 'Green', 'Blue')
)

tube_capacity = 6

generated_goal = (
    ('Red', 'Red', 'Red', 'Red'),
    ('Green', 'Green', 'Green', 'Green'),
    ('Blue', 'Blue', 'Blue', 'Blue')
)

generated_h(initial_state) = 28
returned_cost = 25
```

```
optimal_cost = 17
```

The generated A* implementation hard-codes one color-to-tube assignment, whereas the SearchBench goal accepts any assignment in which every tube is monochromatic. This incorrectly restricts the set of recognized goal states. For this particular instance, however, both the generated and reference solutions happen to reach the same color-to-tube assignment. The direct cause of the cost gap is therefore the generated heuristic, which assigns the initial state a value of 28 although the true remaining cost is 17. The heuristic is inadmissible and removes A*'s optimality guarantee. The returned solution is correct but has cost $25/17 = 1.47$ times the optimum; the error is in the goal and heuristic encoding rather than path recording.

### D.2 Water Jug

**Problem statement.** The instance contains six labeled water jugs with capacities $41, 26, 44, 75, 40$, and $136$ liters and three initially empty buckets. The buckets must contain $274, 297$, and $343$ liters, respectively. At every step, their amounts must remain in non-decreasing order, so the amount in each bucket cannot exceed the amount in the bucket after it. An action (`'+'`, `X`, `Y`) adds $X$ liters to bucket $Y$, while (`'-'`, `X`, `Y`) removes $X$ liters. A bucket cannot be overfilled or reduced below zero. The objective is to reach the target amounts using the minimum number of actions.

```
initial_state = (0, 0, 0)

jug_capacities = [41, 26, 44, 75, 40, 136]
goal_state = (274, 297, 343)

generated_h(initial_state) = 914
returned_cost = 31
optimal_cost = 15
```

The generated heuristic sums the raw volume differences between the current and target amounts. It therefore assigns the initial state a value of 914, although the true remaining cost is only 15 actions. This overestimate makes the heuristic inadmissible. Moreover, the optimal strategy fills the buckets from last to first, which preserves the ordering constraint while effectively decomposing the problem into three simpler subproblems. The generated implementation instead searches over all buckets jointly and returns a correct but suboptimal 31-action solution. Its cost is $31/15 = 2.07$ times the optimum. The error is therefore in the heuristic and search strategy rather than in feasibility, path recording, or cost accumulation.

## E Compute Time of LLM-Generated Codes

In this section, we analyze the computation time of programs generated by LLMs that produce correct solutions. We compare this time to the duration required to calculate the optimal solution for the problem instance using our fast A* implementation. This comparison provides insights into the efficiency of the algorithms generated by the LLMs. The average compute time of LLM-generated codes, normalized against the compute time of our A* implementation for the given instance, is reported in Fig. 8.

Our findings indicate that LLM-generated implementations are significantly slower than our A* implementation. Specifically, GPT4's A* implementations were 213 times slower than the optimal A* solution, suggesting that GPT4's heuristics are still less efficient. Additionally, on average, GPT4's 0-shot code generations that return a correct solution run 900 times slower than the optimal A* implementation. These results underscore the intrinsic difficulty of SearchBench problems, even when addressed through code generation.

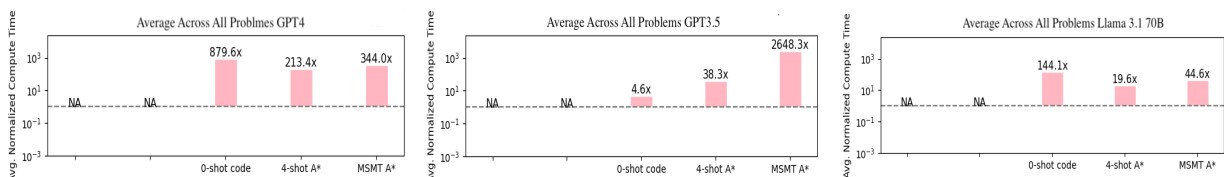

Figure 8: Average compute time of codes returning a correct solution normalized against the compute time of our A* implementation for all problems using GPT4, GPT3.5, Llama 3.1 70B.

## F  SearchBench Variables

Table 4: This table provides a description of each column in SearchBench. Each row in SeacrhBench is an specific problem instance, and columns are fields of each instance.

| Variables | |
| --- | --- |
| **diff_sorted_id** | A unique numeric identifier assigned to each problem instance within a specific problem type. These identifiers are ordered by difficulty level, that is the problem instance with diff_sorted_id of 1 is easier than the instance with diff_sorted_id of 50. |
| **problem_statement** | A natural language description that outlines the problem to be solved. The problem statement is the sole piece of information given to language models when they are instructed to solve SearchBench problems. |
| **problem_type** | Indicates the problem type, out of 11 problem types in SearchBench, that this particular problem is an instance of. |
| **problem_category** | The specific category, out of the five predefined problem categories in SearchBench, to which this problem belongs. |
| **relative_diff_score** | A numeric score that indicates the difficulty of this problem instance relative to other instances within the same problem type. This value is not comparable across different problem types. |
| **opt_solution** | A list of actions that, starting from the given initial state, lead to the goal state with the minimum cost as defined by the problem's criteria. |
| **opt_solution_cost** | The cost of the optimal solution for this problem instance. |
| **opt_solution_compute_t** | The time, in seconds, that our instance-agnostic A* implementation for the problem type took to solve this specific problem instance. |
| **solution_depth** | The number of actions required to reach the goal state from the given initial state with the minimum cost. Together with an upper bound $b$ on the branching factor, a solution depth $d$ gives the coarse tree-size upper bound $\sum_{i=0}^{d} b^i = O(b^d)$ before duplicate-state merging. |
| **max_successor_states** | The maximum number of successor states that can be reached from any given state in this problem. This value is an upper bound on the branching factor of the state search tree for this problem. |
| **num_vars_per_state** | An upper bound on the number of variables in each state of the problem. This value characterizes the size of an individual state representation; total search memory also depends on the number of stored states. |
| **is_feasible_args** | A list of variables of the problem instance that must be passed to the 'is_feasible' function of the evaluation pipeline to determine whether a suggested solution adheres to the rules and constraints of the problem. |
| **is_correct_args** | A list of variables in the problem statement of this instance that must be passed as arguments to the 'is_correct' function in the evaluation pipeline, in order to evaluate the correctness of a suggested solution. |
| **A*_args** | Variables of this problem instance that must be passed to our A* implementation for the problem type to obtain the optimal solution for the instance. |

## G  Search Tree Size Analysis

Figure 9 presents the relationship between the size of the state search tree and the difficulty levels of instances in SearchBench. It displays the average solution-depth and max_successor_states (normalized against the

Table 5: Statistics of metrics pertaining to the search-tree size of a specific instance, compared across all instances within SearchBench.

| Statistics | | | | | | | |
|---|---|---|---|---|---|---|---|
| name | type | min | median | max | mean | standard deviation | missing |
| opt_solution_compute_t | float (seconds) | 0.018 | 0.068 | 599.044 | 17.363 | 67.513 | 0% |
| solution_depth | int | 4 | 14 | 46 | 15.516 | 7.89 | 0% |
| max_successor_states | int | 4 | 12 | 132 | 24.633 | 24.622 | 0% |
| num_vars_per_state | int | 2 | 13 | 60 | 14.785 | 12.05 | 0% |

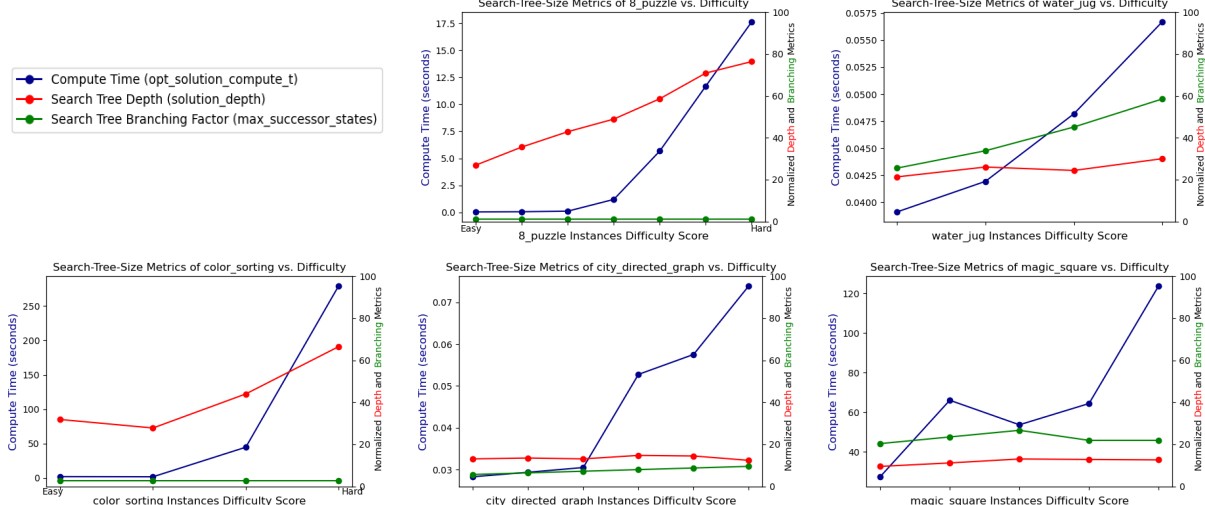

Figure 9: The plots depict the correlation between the increasing difficulty level and the corresponding increase in three metrics: the average depth of the solutions, the branching factor of the state search tree, and the rapid growth of the time required by our A* algorithm to solve the instances, demonstrated across five problem types in SearchBench.

maximum and minimum solution_depth and max_successor_states across all instances in SearchBench) for one problem type from each of the five categories in SearchBench. Additionally, it shows the time our A* algorithm took to navigate the search tree for instances of variable difficulty (compute time is averaged across instances with the same difficulty). We used a machine with 96 64-bit Intel Xeon Gold 5220R CPUs with a maximum speed of 4GHz, and 71.5 MiB Level 3 cache to run the A* implementations.

The figure shows that the solution depth increases linearly with the difficulty scores of problem instances. However, for the city graph, it remains relatively constant, suggesting that the optimal number of hops to reach a destination node from a start node is consistent for our chosen range of directed graph connectivity and sizes (10 to 15 nodes). The max_successor_states, which represents the upper bound on the number of actions leading to successor states from each state, either remains constant or grows linearly with increasing difficulty level. This metric upper-bounds the branching factor of the search tree.

However, the compute time required to navigate the search tree grows rapidly for most problems over the sampled difficulty range. For a search tree with branching factor at most $b$ and solution depth $d$, the number of generated nodes through depth $d$ is bounded by $\sum_{i=0}^{d} b^i = O(b^d)$ before duplicate-state merging. A* may

expand fewer states when its heuristic is informative, although its worst-case time and memory requirements can remain exponential. In tree search, BFS requires $O(b^d)$ time and memory, whereas DFS requires $O(bm)$ memory and $O(b^m)$ worst-case time for maximum depth $m$, and does not generally guarantee a minimum-cost solution (Russell & Norvig, 2020). In our experiments, a BFS implementation did not finish executing even for some of the easiest instances within a 12-hour window.

## H    Broader Impact

Our research, which aims to assist the development of models capable of general reasoning and reliable problem-solving, has the potential to yield significant societal benefits. Combinatorial problems, like those in our dataset, are fundamental in fields such as robotics, logistics, network design, and industrial optimization. Developing models that can tackle unique versions of these problems by designing efficient algorithms or performing systematic searches end-to-end could greatly enhance AI's applicability across various domains. However, this improvement in the reasoning capabilities of language models could also lead to job displacement, as these models could increasingly automate complex tasks traditionally performed by humans.

## I    Hosting, Licensing, and Maintenance

We accept responsibility for any violations of rights that might have occurred in the curation of this dataset. We affirm that the dataset is composed solely of search problems and does not include any sensitive information. The data and code associated with SearchBench are licensed under the Creative Commons (CC BY-SA) license, ensuring open access and usability for the research community.

To ensure the long-term availability and preservation of the SearchBench dataset, we have hosted it on both Hugging Face and GitHub. Moreover, we will provide full access to the code for prompting and inference methods, as well as automated pipelines for generating and evaluating an arbitrary number of instances through these platforms, after the double blind review period. We are committed to maintaining the dataset on these platforms with continued open access. Additionally, we anticipate releasing future versions of this dataset with increased scalability.

