# OpenReview forum: "Navigating the Labyrinth: Evaluating LLMs’ Ability to Reason About Search Problems"
_TMLR — Decision pending for TMLR_

### Review · Reviewer_VF53 · 2025-11-02

**Summary Of Contributions:**

The submission introduces SearchBench, a benchmark of 5 categories and 11 problem types designed to stress test search and backtracking in LLMs. Instances are automatically generated and paired with an evaluation pipeline that scores model outputs along feasibility, correctness, and optimality, with optimal solutions computed via an A* implementation. The fixed benchmark comprises 1,107 instances of varying difficulty. The paper also proposes A* prompting and Multi‑Stage‑Multi‑Try that gates code with simple unit tests and evaluates a range of models under several prompting strategies.

Strength:

1. This paper presents clear problem taxonomy and rule modifications that attempt to avoid pretraining leakage and contamination. Also, the proposed MSMT A* is a simple, pragmatic inference recipe that measurably improves reliability

2. Multi‑dimensional evaluation (i.e. feasible, correct, optimal) is well‑motivated and helpful for diagnosing partial progress.

3. Error and ablation analyses across baselines.

Weakness:

1. MSMT A* uses a multi‑try set-up and unit‑test gating, while other methods appear to use single one-time generations. This makes cross‑method comparisons partly about inference budget and filtering rather than prompting alone. Also, o1 is not evaluated with A* / MSMT due to context limits (page 7), yet figures compares GPT‑4+MSMT to o1.

2. Some outcomes are weird without more analysis, for example, in Color Sorting and Water Jug, MSMT A* attains very high correctness yet 0% optimality, suggesting issues in how costs are encoded or recorded by generated programs and the oracle. The paper mentions such mismatches qualitatively but does not drill down per type.

3. Regarding evaluations, the models used (o1, gpt-4) are mostly released in early 2024 and no sota 2025 models (e.g. gpt‑5,  DeepSeek‑v3.2) have been evaluated. These newer reasoning models have substantial gains in math/coding/logic reasoning, and the gaps of evaluation in the paper may understate current best‑case performance.

4. In addition, while SearchBench itself is useful, the prompt method somehow looks incremental and lacks novelty because leveraging code generation with multi-turn/unit-tested prompting is closely related to well-studied llm agents like in [1] [2] [3] [4]. MSMT largely amounts to prompt engineering + simple gating, the paper conducts no training or fine-tuning to improve the generated code (e.g., via RLVR, self-refine), so the contribution sits more on evaluation and prompt design than on modeling advances.

Typos:  "Trampoline Park" and "Trampoline Matrix" naming differs across the paper; there are spelling typos (e.g., “SeacrhBench” in Table 4 page 17); and a few references rely on Wikipedia for algorithmic properties (page  2, 12, 18)

[1] Gao, Luyu, et al. "Pal: Program-aided language models." International Conference on Machine Learning. PMLR, 2023.

[2] Yao, Shunyu, et al. "React: Synergizing reasoning and acting in language models." The eleventh international conference on learning representations. 2022.

[3] https://github.com/QwenLM/Qwen-Agent

[4] https://github.com/bytedance/SandboxFusion

**Audience:**

Yes

**Audience Explanation:**

The work targets systematic search and backtracking in LLMs, and offers a benchmark with automated generation. Researchers in language model agents and reinforcement learning with verifiable reward may be interested in the paper.

**Claims And Evidence:**

Yes

**Claims Explanation:**

1. The paper asserts that there is a performance gap between text-only reasoning and code generation in combinatorial search problems. This claim is supported by the evaluation result on the SearchBench (containing 1107 synthetically generated puzzle instance) by multiple models under the text-only, A* code generation, and MSMT A* code generation setup. Specifically, with MSMT A* code generation, GPT‑4 achieves 57.1% correct and 28.6% optimal on average across problems, significantly outperforms the text-only baseline (0.6% correct and 0.2% optimal).
.

**Requested Changes:**

1. Explain the correct but not optimal gaps per type with examples. In particular, for the color sorting and water jug cases, please analyze why MSMT A* achieves high correctness yet 0% optimality in Table 3 by including concrete examples (input + output) showing where costs diverge between generated code and the oracle, and whether the issue is cost encoding / heuristic / path recording / etc.

---

> ### Author Response · Authors · 2026-03-07
>
> Thank you for the thorough review. We address each point below.
>
> **On MSMT inference budget and comparison to o1.** We would like to note that MSMT is not a majority-vote approach. The solution returned by the first model-generated code that passes the unit tests is taken as the final answer, requiring only ~1.5x inferences on average vs. 5x–100x in majority-vote methods (Sec. 3). Regarding o1: o1 internalizes inference-time search through extended chain-of-thought reasoning and additional thinking tokens. The GPT-4+MSMT vs. o1 comparison is between two distinct approaches to scaling inference-time compute: external multiple-try attempts vs. internalized reasoning. We will make sure to clarify this comparison in the revision.
>
> **On correct but not optimal solutions in Color Sorting and Water Jug (Table 3).** This pattern is a diagnostic finding in Table 3, and we provide the requested concrete analysis below. In both cases, the models correctly capture the problem's structural rules (achieving high correctness) but fail to properly encode the modified cost semantics specific to SearchBench, including designing admissible heuristics necessary for finding an optimal solution in A\*. This is because they default to classical formulations of these problems and fail to generalize to the rule modifications in SearchBench.
>
> *Color Sorting (91% correct, 0% optimal):* In the Color Sorting problem, the player is given several tubes, each filled with balls of different colors, and must rearrange the balls by moving one ball at a time from the top of one tube to the top of another, until every tube contains balls of only a single color. In the classical version of this puzzle, each color is pre-assigned to a specific tube, giving the solver a single fixed goal state. In SearchBench's modified version, there is no prescribed color-to-tube mapping: the goal is simply that each tube ends up with balls of one color, regardless of which color goes where. This means there are multiple valid goal states (for 3 colors, there are 3!=6 possible assignments), and an optimal solver should search over all of them.
>
> The root cause of the 0% optimality is *goal state encoding*. GPT-4's generated A\* hardcodes a single fixed color-to-tube assignment as its goal, e.g., `goal_state = (('Red','Red','Red','Red'), ('Green','Green','Green','Green'), ('Blue','Blue','Blue','Blue'))`. This restricts the search to just one of the six valid goal arrangements. Our oracle A\* instead uses the flexible condition `all(len(set(tube)) <= 1 for tube in state)`, which accepts any valid assignment. On instance 1 (3 tubes, 4 balls each, capacity 6), the LLM finds a 25-move path to its hardcoded goal, while the optimal solution requires only 17 moves to a different color-tube assignment. The model defaults to the classical fixed-goal formulation rather than SearchBench's modified, more flexible one. This aligns with Table 2's finding that "Encoding Goal State" has only 20% accuracy in puzzle problems.
>
> *Water Jug (95% correct, 0% optimal):* In the classical Water Jug problem, the player has a set of labeled jugs of known capacities and must measure out a specific amount of water into a single container by filling and emptying the jugs. SearchBench modifies this problem in three ways (Fig. 1): (1) instead of one target container, there are multiple buckets that must each be filled to a different target amount; (2) each jug can be used to either add or remove water from any bucket; and (3) a strict ordering constraint is enforced: at every step, the amount of water in each bucket must be less than or equal to the amount in the next bucket (i.e., bucket amounts must remain in non-decreasing order at all times). The cost to minimize is the total number of add/remove actions.
>
> The root cause of the 0% optimality is a combination of an inadmissible heuristic and failure to exploit the ordering constraint. GPT-4 generates the heuristic `h = sum(abs(a - b) for a, b in zip(state, goal))`, which sums the raw volume differences across all buckets. For instance 1 (jugs [41, 26, 44, 75, 40, 136], targets [274, 297, 343]), this returns ~914 for the initial state, while the true optimal cost is only 15 actions, an overestimate of ~60x that violates A\*'s admissibility requirement for optimality guarantees. Furthermore, the optimal strategy exploits the ordering constraint by filling buckets independently from last to first: since the last bucket has the largest target, filling it first and working backwards automatically satisfies the non-decreasing order at every step, effectively decomposing the 3-bucket problem into three independent 1-bucket problems. GPT-4's implementation instead solves all buckets jointly without leveraging this structure, resulting in a correct but highly suboptimal 31-action solution.

---

> > ### Author Response · Authors · 2026-03-07
> >
> > **On novelty of MSMT relative to prior work.** Our primary contribution is the SearchBench benchmark itself, and the finding that iterative computations constitutes a bottleneck for autoregressive models (Sec. 1). MSMT serves as a pragmatic evaluation recipe to establish a baseline, and the systematic comparison across five prompting strategies (Sec. 5, Fig. 3) constitutes our analytical contribution showing the limitations of LLMs in backtracking and search. The references cited (PAL, ReAct, Qwen-Agent, SandboxFusion) address complementary settings, generally program-aided reasoning, agentic tool use, and code execution sandboxing, but none focus on evaluating LLMs in solving *new* combinatorial problems with controllable difficulty and automatic generation and evaluation pipelines.
> >
> > **On typos.** Thank you for raising this to our attention. We will fix "SeacrhBench" in Table 4, standardize "Trampoline Park"/"Trampoline Matrix" naming throughout, and replace Wikipedia citations with primary algorithmic references.
> >
> > Thank you again for reviewing our work. We hope this response answeres your questions.

---

### Review · Reviewer_6Ph9 · 2025-12-20

**Summary Of Contributions:**

The paper introduces SearchBench, a new benchmark designed to evaluate large language models on combinatorial search problems that require backtracking, exploration of multiple paths, and finding optimal solutions. The benchmark includes five problem categories and eleven problem types, with automated pipelines for instance generation and evaluation. Solutions are evaluated along three dimensions: feasibility, correctness, and optimality. A key contribution is the observation that performance improves substantially when models are instead prompted to generate executable search algorithms (in particular A* search). The authors further propose a Multi-Stage-Multi-Try (MSMT) inference strategy that improves the reliability of generated code and leads to gains in performance.

**Additional Comments:**

Overall, this is a nice and easy to read paper. Benchmarks are well designed, the experiments are thorough, and the main insights are clear and important. With minor clarifications regarding scope and positioning, the paper would make a valuable contribution to the literature on reasoning and evaluation of large language models.

**Audience:**

Yes

**Audience Explanation:**

The paper will be of clear interest to researchers working on reasoning, planning, evaluation benchmarks, and the limitations of large language models. It addresses a fundamental question about what kinds of reasoning current LLMs can and cannot perform, and provides a benchmark that is likely to be reused by the community.

**Broader Impact Concerns:**

The paper does not raise significant ethical concerns.

**Claims And Evidence:**

Yes

**Claims Explanation:**

The claims are well supported by extensive experimental evidence. The paper evaluates multiple families of models across several prompting strategies, including  A* prompting, and the proposed MSMT framework. Results are reported consistently across feasibility, correctness, and optimality, which provides a clear and convincing picture of model behavior.

**Requested Changes:**

1. Expand discussion on near-optimal solutions and whether alternative evaluation metrics could provide additional insight.
2. Include a small human baseline or qualitative discussion of human performance on a subset of SearchBench problems.
3. Clarify the scope of what SearchBench measures. In particular, the paper should more explicitly discuss to what extent the strongest results reflect reasoning ability versus offloading reasoning to external code execution.

---

> ### Author Response · Authors · 2026-03-07
>
> Thank you for reviewing our work. We address each point below.
>
> **1. Human baseline.**
>
> Conducting formal human experiments was beyond the scope and resources of our current study. That said, we ground our analysis in established cognitive science findings. Pizlo & Li (2005), discussed in Sec. 2, show that humans can solve the 15-puzzle (a harder variant of our 8-puzzle problems) using simple pyramid-based heuristics, typically producing solutions of ~100-200 moves versus the optimal of <80 moves, indicating high correctness but suboptimal performance. Chronicle et al. (2006) show that human performance on combinatorial optimization problems remains feasible but declines as the state space increases. Additionally, Atwood & Polson (1976) found that humans solve water jug problems effectively. Based on these findings, we designed SearchBench such that problems could be solved correctly but not optimally by humans (i.e. the authors themselves), placing approximate maximum human correctness at ~100%.
>
> **2. Scope: reasoning ability vs. offloading to code execution.**
>
> This is an important distinction and one we see as central to the paper's contribution. Our results show that LLMs struggle with iterative reasoning and backtracking in text (<1% correct with 0-shot text for GPT-4, Fig. 3) but can formalize search strategies as executable code with considerably higher success (57.1% correct with MSMT A*). As discussed in Sec. 7, this contrast suggests that the limitation lies specifically in iterative computation within autoregressive generation, rather than in abstract reasoning about search strategies.
>
> This finding connects to prior work on compositional reasoning limits (Dziri et al., 2024, cited in our paper), which demonstrates that transformers treat multi-step compositional tasks as linearized compute graph matching rather than systematic reasoning, with performance degrading as compositional depth increases. Our work extends this to combinatorial search, showing both where models fail (iterative state space exploration in text) and a practical mechanism (code generation) that leverages models' strength in non-iterative abstract reasoning.
>
>
> **3. Near-optimal solutions and alternative metrics.**
>
> Tab. 3 (Appendix C) provides a per-problem-type breakdown that reveals informative variation. For instance, in 8-Puzzle, MSMT A* achieves 68% correctness and 68% optimality (near-perfect optimization when correct), while in Color Sorting, 91% correctness but 0% optimality, indicating the model correctly sorts but fails to optimize cost under the modified rules (i.e. the model uses the heuristics that are only applicable to the classical versions of the problems). Tab. 2 (Sec. 6) identifies seven specific failure modes across A* subtasks (e.g., encoding goal states at 20% accuracy for puzzles, recording paths at 13% for pathfinding), providing granular diagnostics that complement aggregate metrics.
>
> We hope this response addressed your questions, and we thank you again for reviewing our work.

---

### Review · Reviewer_RXzs · 2026-02-19

**Summary Of Contributions:**

The paper considers reasoning about search problems using LLMs. The authors check three types of qualities in reasoning: feasibility, correctness, and optimality. They primarily focus on LLM reasoning meaning no symbolic reasoning connection/agentic interface is considered. Main results are: a benchmark that groups into four categories: puzzle, subset sum, sorting, undef system, path finding. Furthermore, the authors show that models fail when asked for direct solutions and steps, work much better when asked to produce particular algorithms (A*), best results are observed with Multi-Stage-Multi-Try (MSMT) approach - that they tailor for the considered problems.

**Audience:**

Yes

**Audience Explanation:**

Cannot answer this given the criteria as Machine Learning Research is not my primary research field.
But it looks credible to me.

**Broader Impact Concerns:**

None.

**Claims And Evidence:**

Yes

**Claims Explanation:**

Code, prompts, output, evidence is provided.
However, I have not checked the code, outputs and supplemental material in detail.
Sources are a bit strange as some instances are simply hard-coded and no generic approach is visible, which makes it somewhat useless from a development, understanding, and debugging perspective if each problem instance is hard-coded.

**Requested Changes:**

Overall, I find the work interesting, but I have the following comments / suggestions:
- Problems are not well-defined from a more abstract perspective. Mostly, the authors provide just a vague idea on problems / instances but no indication on how hard the problem and the considered instances are. When taking a look into the source code (just sampling a few), I find them really easy (not almost saying tiny) for standard techniques. While this is probably fine for this type of research, I suggest that the authors to provide more details here. How hard are the problems and considered instances? How to you generate the instances? Is there standard algorithmic solving baseline? Are there known theoretical limitations in this setting? e.g., can it only work if you teach it the right solution approach?
- I find the observation on the performance not really paradox (“fails at a simple task solving step by step, whereas performance improves when prompted to generate a complete A* algorithm”). This is somewhat expected, right? If you give it a particular technique and hints to solve it, it should clearly become better. In addition, if you would have a combinatorial problem and now you tell it that a heuristic is perfectly fine, it becomes easier, or?
- The varying difficulty should really be clarified. While it sounds good to have existence, give solution, enumeration, optimization, it might not tell much depending on the setting.
- The way the problem is phrased might have an influence here already. What is the exact way you are describing it?
- How do you classify the state space?

---

> ### Author Response · Authors · 2026-03-07
>
> Thank you for taking the time to review our work. We address each point below.
>
> **1. Problem definitions, difficulty, and instance hardness.**
>
> The problem framework in Sec. 2 defines each problem in terms of an initial state, a goal state, and a set of possible actions, where the task is to find a minimum-cost action sequence. Difficulty is determined by the state space size of the instance (Sec. 2), and we provide quantitative metrics for every instance in Tab. 4 and Tab. 5 (Appendix E): solution_depth (number of actions in the optimal solution), max_successor_states (upper bound on branching factor), and num_vars_per_state (state representation size). Fig. 9 (Appendix F) shows the correlation between difficulty level and these metrics, including exponential growth in compute time for our A* execution. Tab. 1 provides an instance example with the modified rules highlighted. We will make these connections more prominent in the main text to improve accessibility.
>
> On algorithmic baselines: for each of the 11 problem types, we implemented a fast A* algorithm with a provably admissible and consistent heuristic that produces the unique optimal solution (Sec 2). Because SearchBench modifies the rules of known problems (Fig. 1), standard off-the-shelf algorithms for the classic versions do not directly apply, rather a correct solver must incorporate these modifications. The problems are predominantly NP-hard (Sec. 1), and our categories map to the four classical types in theoretical computer science (Wilson, 2016), as discussed in Sec. 2. We will ensure to foreground these details in the revision.
>
> **2. On the performance observation.**
>
> We agree that generally prompting the model in code improves performance. The observation we highlight is that the cognitively simpler task (solving a puzzle step-by-step, which children and non-experts can do) fails, while the cognitively harder task (writing a complete search algorithm from scratch, typically the level of an undergraduate CS assignment) succeeds. The underlying reason, as we discuss in Sec. 7, is that algorithm generation is a non-iterative task that offloads the exponential state space exploration and the long-horizon backtracking/search from the autoregressive model to code execution . The model's core limitation in solving SearchBench is in iterative computation and backtracking within text, not in abstract reasoning about search strategies.
>
> We also note that if A* prompting were simply providing the answer approach, including in-category examples should dramatically improve performance. However, Fig. 4 (Sec. 6) shows that including a solved example from the same problem category yields only marginal improvement (up to 15 additional instances), suggesting that the model generalizes from the prompt structure rather than reusing a demonstrated solution. The in-context examples are always from different categories than the target problem (Sec. 3).
>
> **3. Varying difficulty and the four problem types.**
>
> The four types (existence, construction, enumeration, optimization) follow the established classification of combinatorial problems in theoretical computer science (Wilson, 2016). Each type determines the nature of the task: existence asks "does a solution exist?", construction asks "find a valid solution," enumeration involves "generate solutions systematically," and optimization requires "find the least-cost solution." Each SearchBench category maps to one of these types (Sec. 2), ensuring breadth of coverage in evaluating combinatorial reasoning. The multi-dimensional evaluation (feasibility, correctness, optimality) in Sec. 2 captures how performance varies across these types. Additional analysis is provided in Fig. 7 (Appendix C) and Fig. 9 (Appendix F).
>
> **4. Problem phrasing.**
>
> Tab. 1 shows the exact problem statement given to the model, with instance-specific components in green and SearchBench rule modifications in orange. All prompts for all five methods are provided in the supplementary material. Problems are described entirely in natural language with no encoding assumptions (Sec. 4).
>
> **5. State space classification.**
>
> Tab. 4 (Appendix E) defines all state space variables, including max_successor_states (branching factor), solution_depth (optimal path length), and num_vars_per_state (state dimensionality). Tab. 5 provides summary statistics across all 1,107 instances, and the state space size for each instance is upper bounded by b^d (Appendix E). Fig. 9 visualizes these relationships across problem types.
>
> Thank you again for reviewing our work. We hope this response answers your questions.

---

### Decision · Action_Editor_44kX · 2026-05-20

**Recommendation:** Accept with minor revision

**Additional Comments:**

While the reviewers appreciate the direction of the work and empirical evaluation, all the reviewers have highlighted the need for improving the discussion of the problems and analysis of the complexity of these problems for standard search algorithms. The Reviewer RXzs has highlighted clear areas for improvement, which should be taken care of before the paper can be published.

**Audience:**

Yes

**Audience Explanation:**

Given lot of interest in reasoning abilities of LLMs, the topic is timely and the benchmark suite might have impact

**Claims And Evidence:**

Yes

**Claims Explanation:**

The paper proposes a new benchmark that is scalable [i.e., the complexity can vary with parameters] and have shown promising empirical evaluation. All the reviewers have pointed out concerns regarding breadth of evaluations but on the balance, the evaluation is perhaps sufficient.